# High prevalence of antimicrobial resistance and multidrug resistance among bacterial isolates from diseased pets: Retrospective laboratory data (2015–2017)

**Nurul Asyiqin Haulisah**, **Latiffah Hassan** *, **Saleh Mohammed Jajere**, **Nur Indah Ahmad**, **Siti Khairani Bejo**

Faculty of Veterinary Medicine, Universiti Putra Malaysia, Serdang, Selangor Darul Ehsan, Malaysia

* latiffah@upm.edu.my

**Data Availability Statement:** All relevant data are within the paper.

## Abstract

Laboratory surveillance and the monitoring of antimicrobial resistance (AMR) trends and patterns among local isolates have been highly effective in providing comprehensive information for public health decision-making. A total of 396 cases along with 449 specimens were received for antibiotic susceptibility testing at a public university veterinary diagnostic laboratory in Malaysia between 2015 and 2017. *Escherichia coli* was the most frequently isolated (n = 101, 13%) bacteria, followed by *Staphylococcus pseudintermedius* (n = 97, 12%) and *Streptococcus canis* (n = 62, 8%). In cats, *S. pseudintermedius* isolates were highly resistant to azithromycin (90%), while the *E. coli* isolates were highly resistant to doxycycline (90%), tetracycline (81%), and cephalexin (75%). About 55% of *S. pseudintermedius* and 82% of *E. coli* were multi-drug resistant (MDR). In dogs, *S. intermedius* isolates were highly resistant to aminoglycosides neomycin (90.9%) and gentamicin (84.6%), and tetracycline (75%). Whereas the *E. coli* isolates were highly resistant to cephalexin (82.1%) and amoxicillin/clavulanic acid (76.5%). MDR was observed in 60% of *S. intermedius* and 72% of *E. coli* from dogs. Generally, the bacterial isolates from cats demonstrated higher levels of resistance to multiple antibiotics compared to those from dogs.

## Introduction

Antimicrobial resistance (AMR) among bacterial isolates from pets has not received as much attention as AMR among bacteria in livestock. However, AMR among bacterial isolates such as MRSA and MDR *E. coli* and *Klebsiella pneumoniae* in pets is an emerging issue of veterinary and public health significance [1]. In addition to AMR complicating the treatment of sick pets, close relationships between pets and their human counterparts can result in the transmission of the resistant pathogens between the two [2, 3]. The transmission of antibiotic-resistant bacteria such as the zoonotic transmission of opportunistic pathogens like *Staphylococcus* spp. between dogs and their owners have been reported in many studies [4, 5]. In fact, several studies have reported that the AMR level is higher among animals living in close association with

**Funding:** This research project was part of a Master's research study undertaken in the Faculty of Veterinary Medicine, Universiti Putra Malaysia, and was funded by the UPM Research Grant number GP – IPB/2019/9676500, awarded to LH. The funders had no role in study design, data collection and analysis, decision to publish, or preparation of the manuscript.

**Competing interests:** The authors have declared that no competing interests exist.

humans than the animals residing in remote areas [6]. Studies have found that pets may acquire resistant pathogens from their owners [7–9].

In the past decade, the use of antibiotics in animals reared for meat has been broadly scrutinized [10, 11]. Multiple reports have also established that the volume and frequency of antibiotics used in livestock production settings are reducing the effectiveness of the antibiotics when used for medicinal purposes in humans [12]. In fact, it is among the major drivers of the increasing level of AMR among bacterial pathogens [13, 14]. In contrast, there have been lesser attention paid to the use of antibiotics in pets and especially, the antibiotic resistance among bacteria from pets. Pets are normally exposed to antibiotics when they are treated for bacterial infections [15], or prevented from secondary bacterial infection during episodes of viral infections [16]. Antibiotic prescriptions often include critically important antibiotic groups used in human medicine including the heavy usage of broad-spectrum agents such as fluoroquinolones, cephalosporins, and penicillins [17]. These antibiotics are categorized as veterinary critically important antibiotics by the OIE [18] as well as critically important for human medicine by the WHO [19]. Veterinarians are often inclined to use broad-spectrum antibiotics as the first-line treatment for bacterial infection due to its convenience [20, 21] and as blanket coverage against a broad range of bacterial diseases. However, such usage increases selective pressure, leading to the development of multidrug resistance in bacteria such as *Escherichia coli*, *Pseudomonas aeruginosa*, and other commensal bacteria in pets [22–24].

Sporadic reports have suggested an increasing level of resistance in important pathogens in cats and dogs [25] and this situation escalates the complexity of treating bacterial infections such as urinary tract infections and cystitis in these species [26–28]. The findings from this study will highlight the significance of AMR among clinically important bacterial pathogens in small animal medicine. In line with Malaysia Strategic Action Plan for AMR (MyAP-AMR) [29] and Global Action Plan of AMR, this study aims to describe the antimicrobial resistance patterns of clinically important bacterial pathogens from diagnostic cases in pets. This study will also determine the differences in resistance profiles between isolates recovered from diseased cats and dogs between 2015 and 2017.

## Material and methods

### Source of data

The design of this study has been described in Haulisah et al., [30]. The data originated from the veterinary diagnostic cases that were received from various veterinary health premises and animal facilities in Peninsular Malaysia by the accredited Bacteriology Laboratory, Faculty of Veterinary Medicine of Universiti Putra Malaysia (UPM) Serdang, Selangor. Diagnostic reports (paper format) of cases from 1 January 2015 to 31 December 2017 were compiled and transferred into WHONET (vers 5.6, Boston, MA). Relevant data compiled included information concerning animal species, clinical history, sex, type of specimens (urine, wound swab, lavage fluid, etc.), clinical signs, isolated bacteria, and findings from antibiotic susceptibility testing (AST).

The laboratory performed culture and identification of received samples using standard microbiological procedures [31]. The antimicrobial sensitivity was performed using the Kirby Bauer's disc diffusion on Mueller Hinton Agar (MHA) method following the Clinical Laboratory Standards Institute (CLSI) guidelines [32]. The AST data was grouped into three categories: Resistant (R), Intermediate (I), and Susceptible (S) using the established clinical breakpoints. The antimicrobial agent tested varied depending on the request of the clinician, type of the organism, and the availability of the antimicrobial agents.

Data received from the 396 cases by the laboratory were entered into Excel 2007 (Microsoft Office, 2007) and then transferred into WHONET 5.6 [33]. Data related to cats and dogs were

analyzed separately. A total of 780 isolates were recovered from the various clinical samples of dogs and cats over a period of three years (2015–2017).

## Data analysis

Descriptive data analysis was performed in WHONET 5.6 as described in WHO [34, 35]. Trends and patterns of tested antibiotics and the multidrug resistance (MDR) levels among the clinical isolates were analyzed based on the species of the pets followed by interspecies comparison. The difference in the level of resistance based on the antimicrobial agents for the 3-year study period was tested using the Chi-square test at 95% confidence intervals using SPSS 22 (IBM, Armonk, NY: IBM Corp.) at $\alpha = 0.05$. Fisher's exact test was applied when the expected numbers per cell were small. For the purpose of this analysis, the isolates categorized as 'intermediate' and 'resistant' were grouped as 'non-susceptible'. In addition, an isolate was marked as 'multi-drug resistant' if it were resistant to at least one agent in three or more classes of antibiotics [36].

## Results

### Descriptive analysis of samples

Between 2015 and 2017, 396 cases with accompanying 449 specimens were received by the laboratory for antibiotic susceptibility testing (AST) (cats n = 271, 68.4% and dogs n = 125, 31.6%). A total of 780 isolates were obtained out of which more than half were recovered from cats (n = 467, 59.9%) and the rest from dogs (n = 313, 40.1%).

A total of 65 bacterial species were identified from the specimens. The most common species were *Escherichia coli* (n = 101, 13%), followed by *Staphylococcus pseudintermedius* (n = 97, 12%), and *Streptococcus canis* (n = 62, 8%). The majority of *Enterococcus* spp. were *Enterococcus faecalis* (n = 48, 72.7%) and the remaining 18 were non-*faecalis Enterococcus* spp. (Fig 1).

### Antimicrobial resistance of clinical isolates from cats

A total of 289 clinical samples were received (Table 1) from which 467 isolates were recovered. *Staphylococcus pseudintermedius* was the most common bacteria from wounds/abscesses (61.2%) and ear infections (13.4%). Overall, *S. pseudintermedius* (n = 67, 14.3%), *E. coli* (n = 65, 13.9%), *S. canis* (n = 44, 9.4%), and *K. pneumoniae* (n = 37, 7.9%) were the most frequently isolated. *Klebsiella pneumoniae* was found to be more commonly isolated from urine (51.4%) while *Escherichia coli*, *Staphylococcus pseudintermedius*, and *Streptococcus canis* were most commonly isolated from wounds/abscess (Table 1). More than 75.8% (354/467) of the bacteria isolated were MDR out of which 20.6% (96/467) were resistant to one or two antibiotic classes and only 3.6% (17/467) were susceptible to all tested antibiotics.

### Antibiogram based on species of bacteria isolated from cats

Antibiogram of the bacteria isolated is presented in Figs 2–5. The MDR levels for the selected bacteria for 2015–2017 are shown in Fig 6.

**Staphylococcus pseudintermedius.** Fig 2 shows that the highest percentage of resistance was against azithromycin (90%; 95% CI = 54.1–99.5), followed by trimethoprim/sulfamethoxazole (57.1%; 95% CI = 29.6–81.2), and amoxicillin (53.3%; 95% CI = 27.4–77.7). There was no significant change in resistance trends for all tested antibiotics. More than 40.3% (27/67) of the *S. pseudintermedius* were resistant to multiple classes of antibiotics (MDR) (Fig 6).

**Escherichia coli.** The proportion of *E. coli* isolates resistant to doxycycline was 90% (95% CI = 54.1–99.5), 81% to tetracycline (95% CI = 57.5–93.7), and 75% to cephalexin (95%

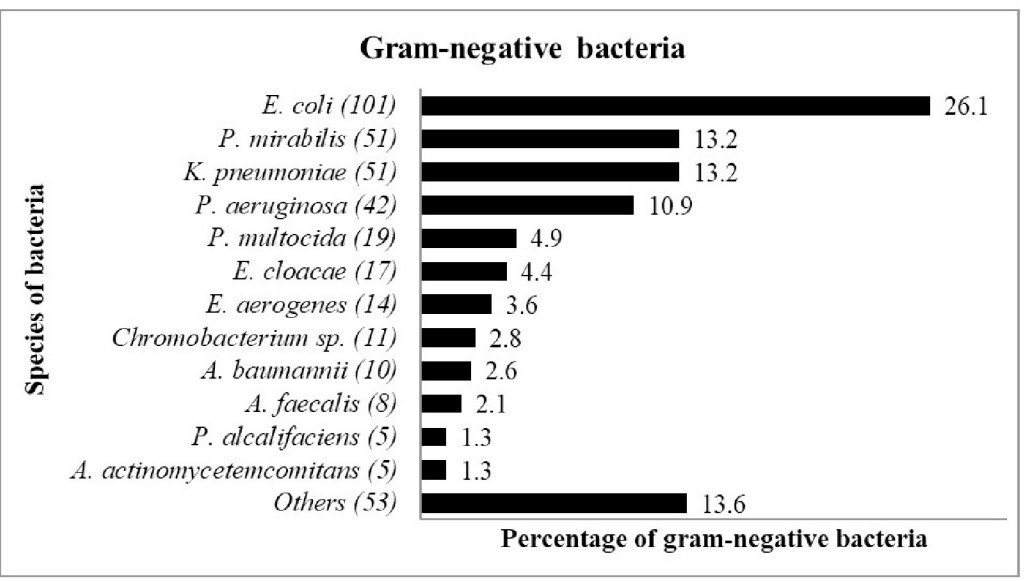

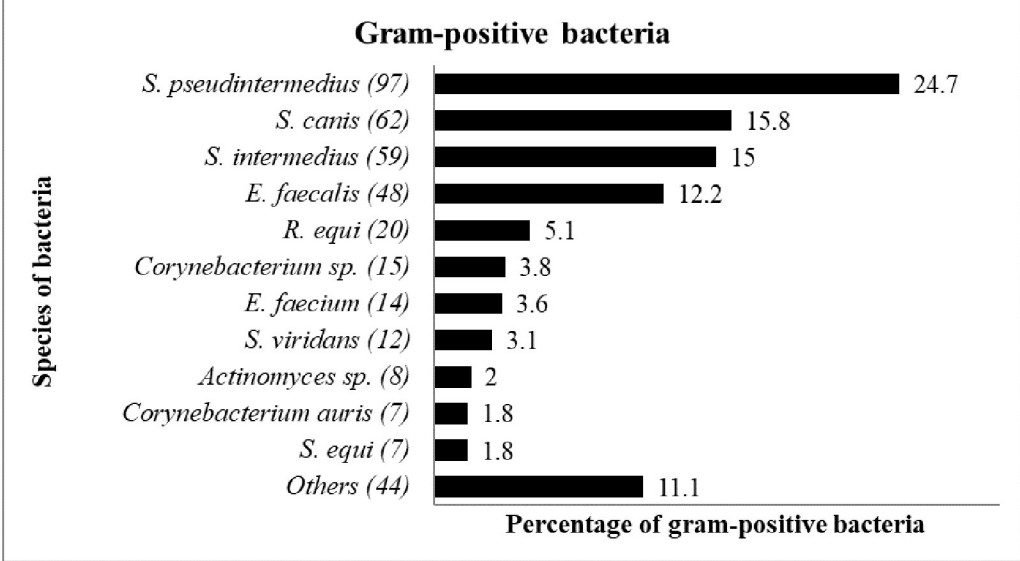

**Fig 1. The distribution of bacterial species (Gram-negative and gram-positive bacteria) from diseased pets isolated between January 2015 and December 2017.** Numbers inside brackets '()' indicate the total number of bacterial isolates.

CI = 60.1–85.9) (Fig 3). There was no significant change in resistance level over the three-year study period. MDR (Fig 6) was observed among 70.8% (46/65) of *E. coli* isolates.

**Streptococcus canis.** Fig 4 shows that the resistance levels were high for gentamicin (66.7%; 95% CI = 35.5–88.7) and enrofloxacin (58.8%; 95% CI = 40.8–74.9). A significant increase in isolates grouped as 'intermediate' was observed for amoxicillin/clavulanic acid (*p* = 0.000, from 13.3% in 2015 to 22.2% in 2017). However, *S. canis* remained highly suscepti-ble (90.7%) to amoxicillin/clavulanic acid over the three years. The MDR of *S. canis* (Fig 6) was relatively low at 9.1% (4/44).

**Klebsiella pneumoniae.** It was found that *K. pneumoniae* isolates were highly resistant to fluoroquinolones marbofloxacin (87.5%; 95% CI = 70.1–95.9), enrofloxacin (82.1; 95% CI = 62.4–93.2), and amoxicillin/clavulanic acid (81.1%; 95% CI = 64.3–91.5) (Fig 5). No

**Table 1. Species of bacteria isolated from diagnostic samples from diseased cats received by the Bacteriology Laboratory between 2015 and 2017.**

| Samples | N | %*Staphylococcus pseudintermedius* (n = 67) | %*Escherichia coli* (n = 65) | %*Streptococcus canis* (n = 44) | %*Klebsiella pneumoniae* (n = 37) | % Others* (n = 254) |
|---|---|---|---|---|---|---|
| Wounds/ abscess | 157 | 41 (61.2%) | 34 (52.3%) | 31 (70.5%) | 11 (29.7%) | 148 (58.3%) |
| Ear | 24 | 9 (13.4%) | 4 (6.2%) | 5 (11.4%) | 1 (2.7%) | 24 (9.4%) |
| Urinary tract | 60 | 6 (9%) | 16 (24.6%) | 1 (2.3%) | 19 (51.4%) | 44 (17.3%) |
| Reproductive | 11 | 3 (4.5) | 5 (7.7%) | 1 (2.3%) | 0 | 6 (2.4%) |
| Post-mortem | 9 | 1 (1.5%) | 0 | 3 (6.8%) | 5 (13.5%) | 5 (2%) |
| Feces | 5 | 1 (1.5%) | 5 (7.7%) | 0 | 1 (2.7%) | 4 (1.6%) |
| Others | 23 | 6 (9%) | 1 (1.5%) | 3 (6.8%) | 0 | 23 (9.1%) |
| **Total** | **289** | **67 (100%)** | **65 (100%)** | **44 (100%)** | **37 (100%)** | **254 (100%)** |

*N*- Total number of clinical samples; n- Total number of bacterial isolates

*Others include blood, respiratory and eye infections

significant change was observed in the resistance trends for all the tested antimicrobial agents over the three years. MDR was found in 86.5% of *K. pneumoniae* isolates from cats (86.5%, 32/ 37). None of the isolates was susceptible to all the classes of tested antimicrobial agents (Fig 6).

## Antimicrobial resistance patterns of clinical isolates from dogs

Between 2015 and 2017, 160 clinical samples were analyzed from a total of 125 cases (Table 2). From these samples, 313 isolates were recovered. *Staphylococcus intermedius* (n = 37, 11.8%), *Escherichia coli* (n = 36, 11.5%), *Proteus mirabilis* (n = 31, 9.9%), and *Staphylococcus pseudintermedius* (n = 30, 9.58%) were the most frequently isolated bacterial species (Table 2). MDR was identified among 71.6% (224 /313) of the isolated bacteria.

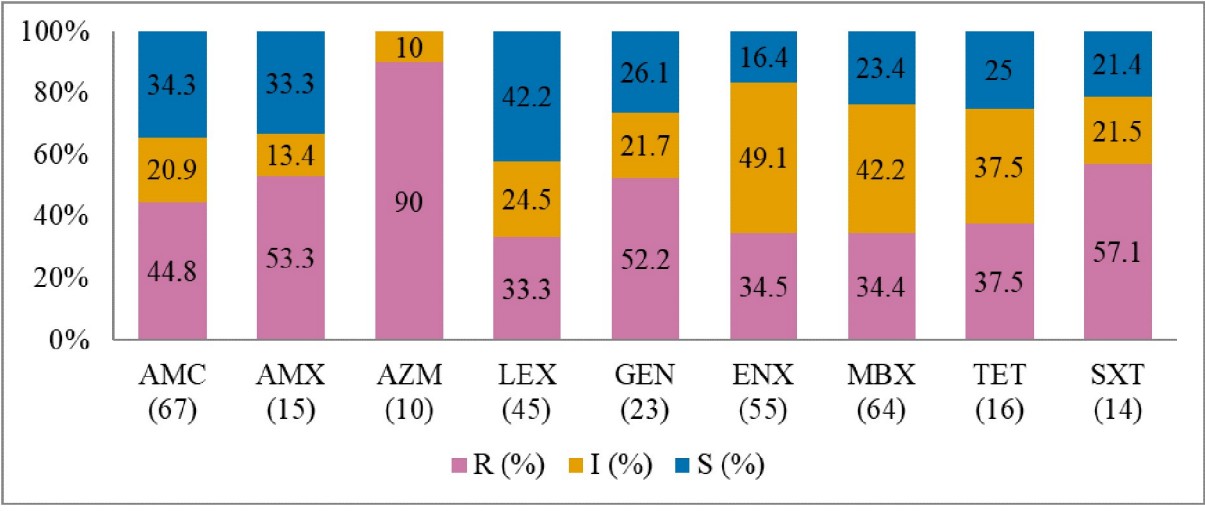

**Fig 2. Antibiotic susceptibility pattern from *Staphylococcus pseudintermedius* isolates from diseased cats (January 2015–December 2017).** R–resistance; I-intermediate; S–susceptible; AMC–amoxicillin/clavulanic acid; AMX–amoxicillin; AZM–azithromycin; LEX–cephalexin; GEN–gentamicin; ENX–enrofloxacin; MBX–marbofloxacin; TET–tetracycline; SXT–trimethoprim-sulfamethoxazole. Numbers inside brackets '()' indicate the total number of tested isolates for each antibiotic.

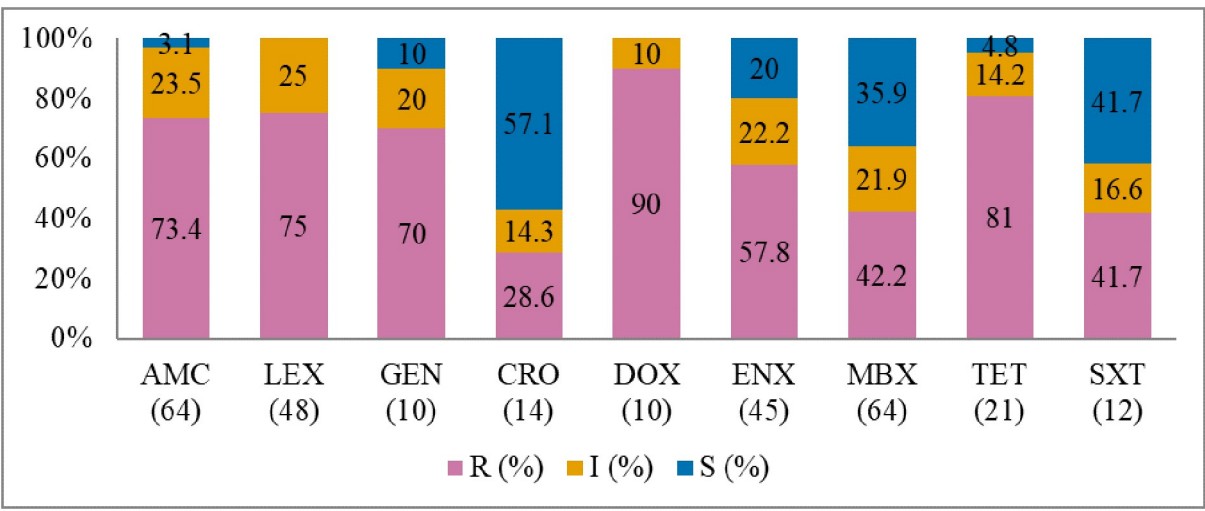

**Fig 3. Antibiotic susceptibility pattern from *Escherichia coli* isolates from diseased cats (January 2015–December 2017).** R–resistance; I–intermediate; S–susceptible; AMC–amoxicillin/clavulanic acid; LEX–cephalexin; GEN–gentamicin; CRO–ceftriaxone; DOX–doxycycline; ENX–enrofloxacin; MBX–marbofloxacin; TET–tetracycline; SXT–trimethoprim-sulfamethoxazole. Numbers inside brackets '()' indicate the total number of tested isolates for each antibiotic.

## Antibiogram based on the species of bacteria isolated from dogs

Antibiogram of the bacteria isolated from dogs is presented in Figs 7–10. The MDR levels for the selected bacterial species (2015–2017) are shown in Fig 11.

**Staphylococcus intermedius.** The AST results for *S. intermedius* are presented in Fig 7. The highest resistance level was against neomycin (90.9%; 95% CI = 57.1–99.5), gentamicin (84.6%; 95% CI = 53.6–97.3), and tetracycline (75%; 95% CI = 47.4–91.7). It was also noted that about 60% (22/37) of the *S. intermedius* isolates were MDR (Fig 11). The resistance trend for the bacteria did not change significantly over time (2015–2017).

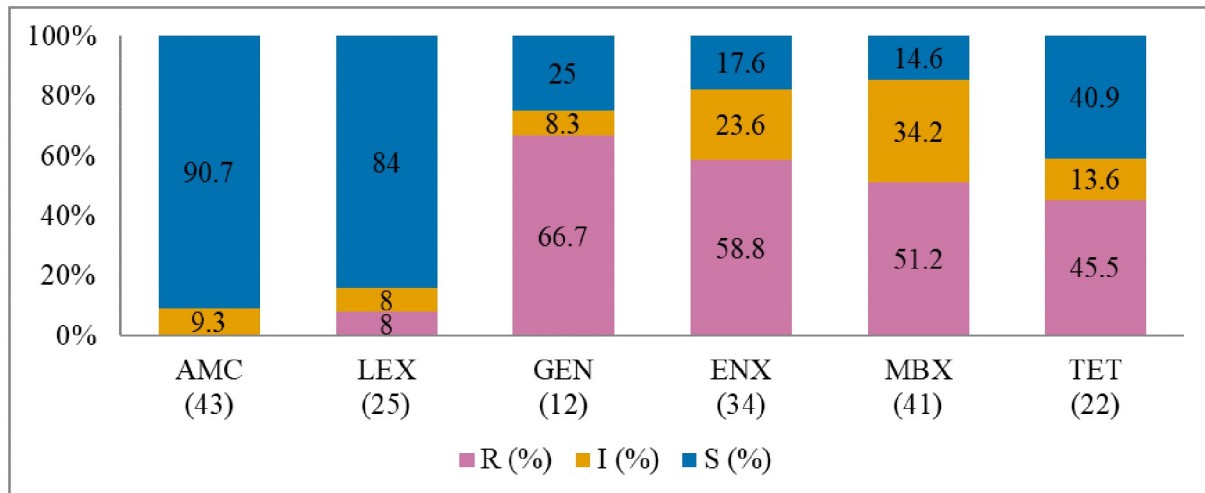

**Fig 4. Antibiotic susceptibility pattern from *Streptococcus canis* isolates from diseased cats (January 2015–December 2017).** R–resistance; I–intermediate; S–susceptible; AMC–amoxicillin/clavulanic acid; LEX–cephalexin; GEN–gentamicin; ENX–enrofloxacin; MBX–marbofloxacin; TET–tetracycline. Numbers inside brackets '()' indicate the total number of tested isolates for each antibiotic.

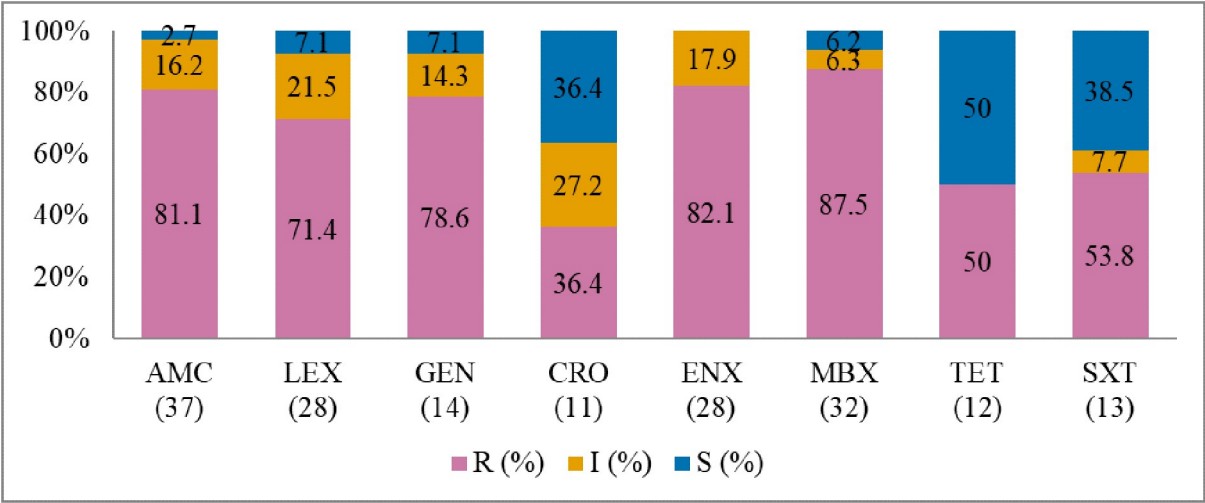

**Fig 5. Antibiotic susceptibility pattern from *Klebsiella pneumoniae* isolates from diseased cats (January 2015–December 2017).** R–resistance; I–intermediate; S–susceptible; AMC–amoxicillin/clavulanic acid; LEX–cephalexin; GEN–gentamicin; CRO–ceftriaxone; ENX–enrofloxacin; MBX–marbofloxacin; TET–tetracycline; SXT–trimethoprim-sulfamethoxazole. Numbers inside brackets '()' indicate the total number of tested isolates for each antibiotic.

**Escherichia coli.** Fig 8 depicts the AMR levels of *E. coli* (2015–2017). The resistance was the highest for cephalexin (82.1%; 95% CI = 62.4–93.2) and amoxicillin/clavulanic acid (76.5%; 95% CI = 58.5–88.6). The resistance trend for *E. coli* did not change considerably over the study period. It was also noted that the proportion of MDR (Fig 11) among *E. coli* was 72.2% (26/36).

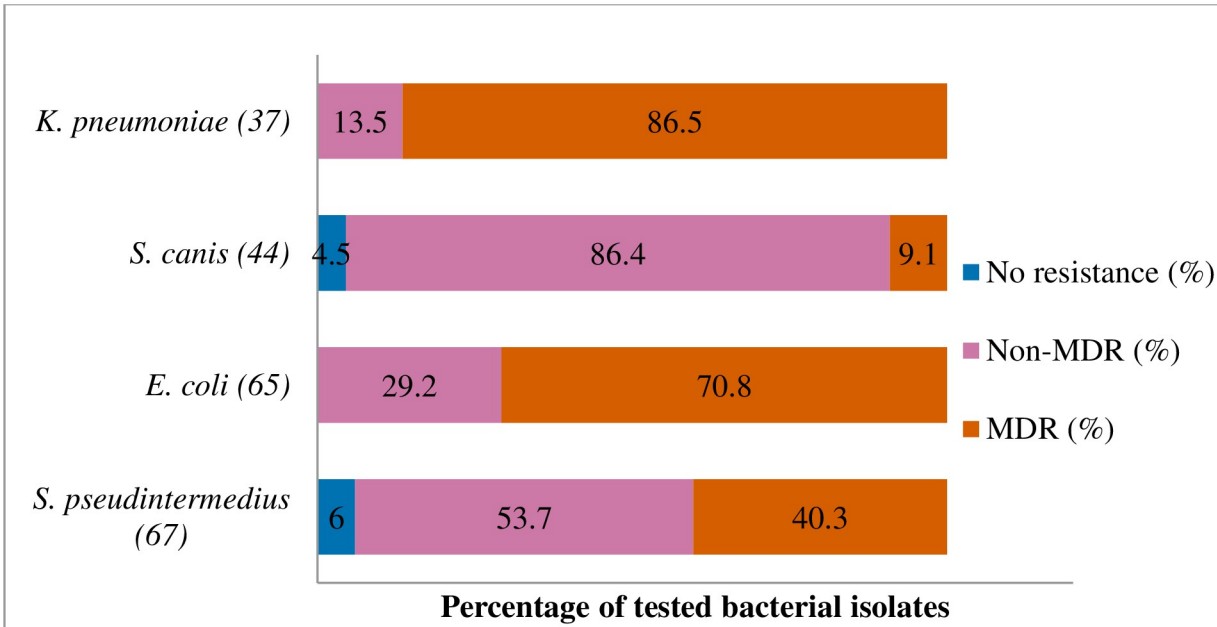

**Fig 6. Multidrug resistance of clinically important bacterial pathogens from diseased cats between 2015 and 2017.** Numbers inside brackets '()' indicate the number of isolates detected; those on bars indicate percentage per organism; Non-MDR–Resistant to 1 or 2 classes; MDR–multi-drug-resistant, resistant to 3 or more classes.

**Table 2. Species of bacteria isolated from diagnostic samples from diseased dogs received by the Bacteriology Laboratory between 2015 and 2017.**

| Samples | N | %*Staphylococcus intermedius* (n = 37) | %*Escherichia coli* (n = 36) | %*Proteus mirabilis* (n = 31) | %*Staphylococcus pseudintermedius* (n = 30) | %Others* (n = 179) |
|---|---|---|---|---|---|---|
| Wounds/abscess | 55 | 20 (54.1%) | 15 (41.7%) | 8 (25.8%) | 8 (26.7%) | 102 (57%) |
| Ear infection | 45 | 13 (35.1%) | 2 (5.6%) | 14 (45.2%) | 13 (43.3%) | 47 (26.3%) |
| Urinary tract | 36 | 2 (5.4%) | 10 (27.8%) | 8 (25.8%) | 7 (23.3%) | 15 (8.4%) |
| Reproductive | 14 | 0 | 9 (25%) | 0 | 0 | 5 (2.8%) |
| Post-mortem | 2 | 0 | 0 | 0 | 0 | 2 (1.1%) |
| Others | 8 | 2 (5.4%) | 0 | 1 (3.2%) | 2 (6.7%) | 8 (4.5%) |
| **Total** | **160** | **37 (100%)** | **36 (100%)** | **31 (100%)** | **30 (100%)** | **179 (100%)** |

*N*- Total number of clinical samples; n- Total number of bacterial isolates.

*Others include blood, respiratory and eye infections

**Proteus mirabilis.** The frequency of AMR of *P. mirabilis* to all the tested antimicrobial agents (2015–2017) is presented in Fig 9. *P. mirabilis* isolates were completely resistant to tetracycline (100%; 95% CI = 59.8–100), followed by cephalexin (79.2%; 95% CI = 57.3–92.1), and trimethoprim/sulfamethoxazole (75%; 95% CI = 35.6–95.5). A significant decrease in resistance level was observed for marbofloxacin (*p* = 0.013, from 87.5% in 2015 to 36.4% in 2017). The MDR of *P. mirabilis* (Fig 11) was 51.6% (16/31).

**Staphylococcus pseudintermedius.** The resistance level was highest for tetracycline at 54.5% (95% CI = 24.5–81.8) (Fig 10). A significantly decreased resistance level to cephalexin (from 53.9% in 2015 to 0% in 2017) and to tetracycline (from 100% in 2015 to 0% in 2017) was observed over the three-year study period. Forty percent (12/30) of the *S. pseudintermedius* isolates were MDR (Fig 11).

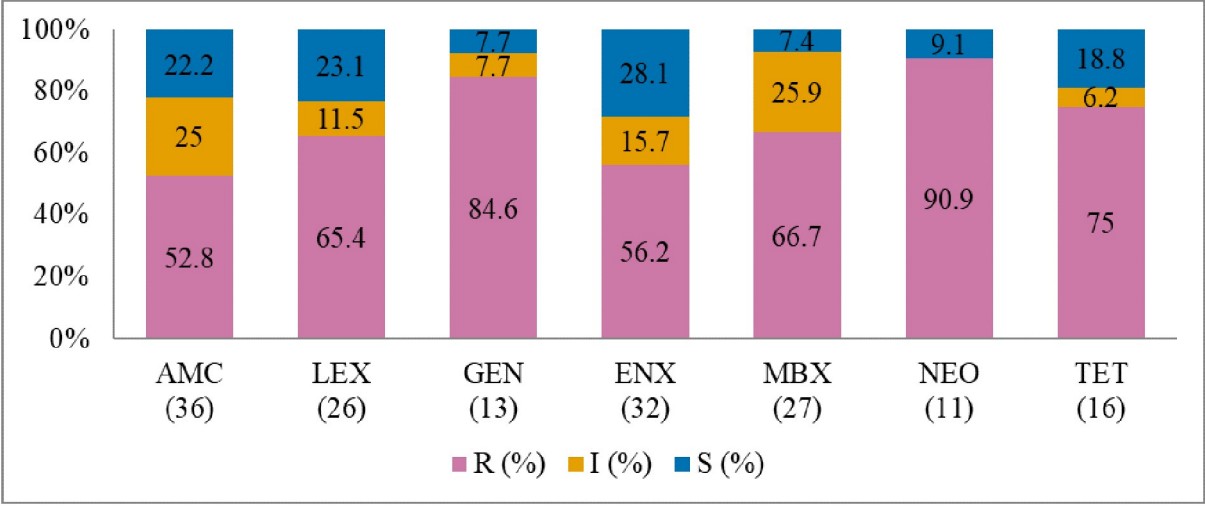

**Fig 7. Antibiotic susceptibility pattern from *Staphylococcus intermedius* isolates from diseased dogs (January 2015-December 2017).** R–resistance; I-intermediate; S–susceptible; AMC–amoxicillin/clavulanic acid; LEX–cephalexin; GEN–gentamicin; ENX–enrofloxacin; MBX–marbofloxacin; NEO-neomycin; TET–tetracycline. Numbers inside brackets '()' indicate the total number of tested isolates for each antibiotic.

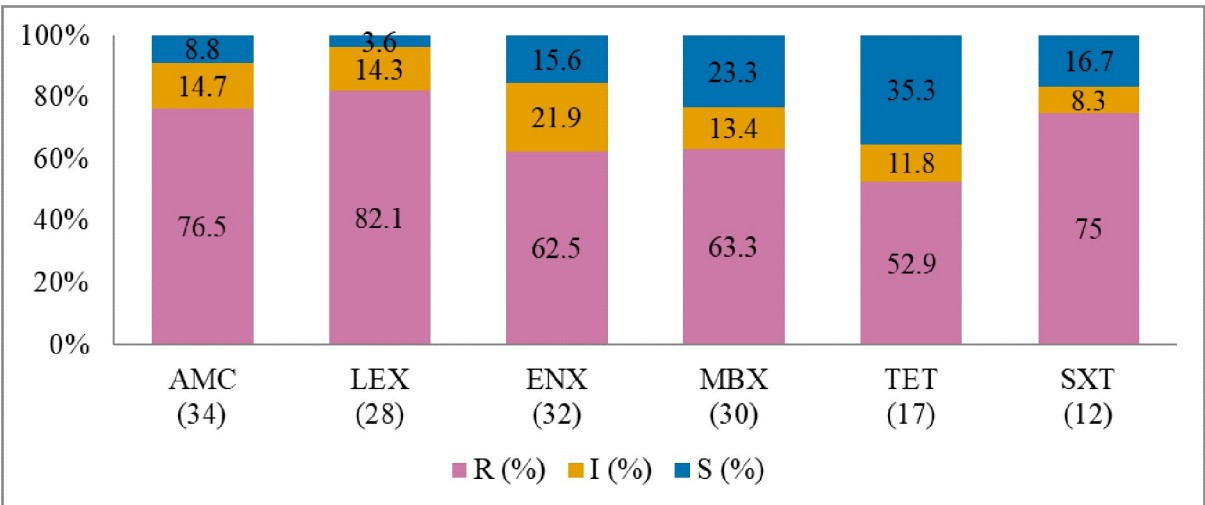

**Fig 8. Antibiotic susceptibility pattern from *Escherichia coli* isolates from diseased dogs (January 2015-December 2017).** R–resistance; I-intermediate; S–susceptible; AMC–amoxicillin/clavulanic acid; LEX–cephalexin; ENX–enrofloxacin; MBX–marbofloxacin; TET–tetracycline; SXT–trimethoprim-sulfamethoxazole. Numbers inside brackets '()' indicate the total number of tested isolates for each antibiotic.

## Differences between the profile of resistances of isolates from cats and dogs

*Staphylococcus pseudintermedius* and *Escherichia coli* were the most commonly encountered bacterial species from both cats and dogs. In addition, most antibiotic classes used to treat *E. coli* and *S. pseudintermedius* infections are shared between these two species. Comparison of the AMR profiles among *S. pseudintermedius* and *E. coli* isolates are presented in Tables 3 and 4 respectively.

**Escherichia coli.** *E. coli* isolates from dogs showed high levels of non-susceptibility towards all the tested antimicrobial agents with higher levels to trimethoprim/

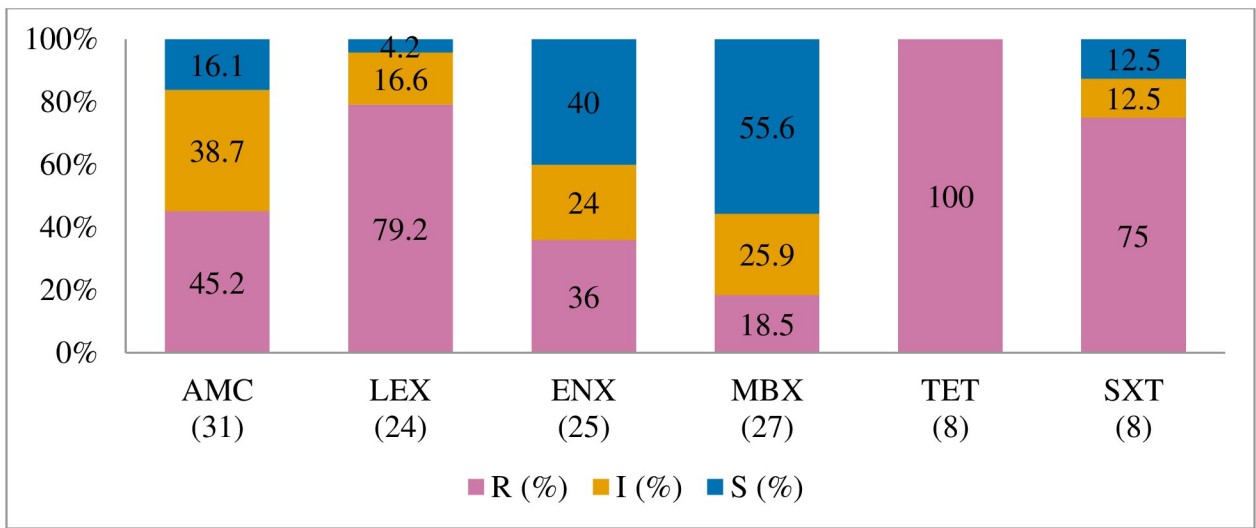

**Fig 9. Antibiotic susceptibility pattern from *Proteus mirabilis* isolates from diseased dogs (January 2015-December 2017).** R–resistance; I-intermediate; S–susceptible; AMC–amoxicillin/clavulanic acid; LEX–cephalexin; ENX–enrofloxacin; MBX–marbofloxacin; TET–tetracycline; SXT–trimethoprim-sulfamethoxazole. Numbers inside brackets '()' indicate the total number of tested isolates for each antibiotic.

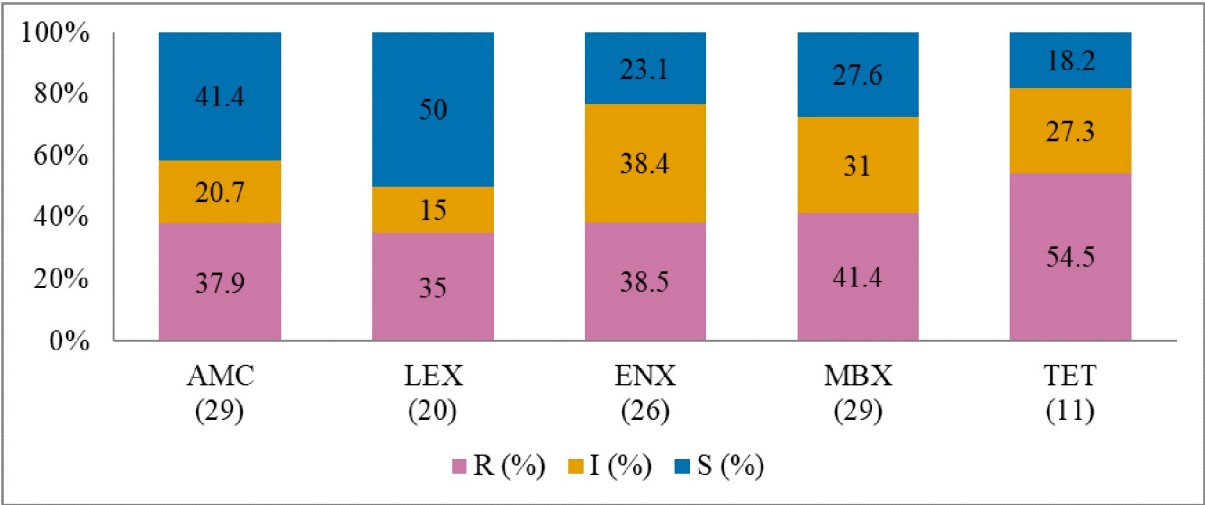

**Fig 10. Antibiotic susceptibility pattern from _Staphylococcus pseudintermedius_ isolates from diseased dogs (January 2015-December 2017).** R–resistance; I–intermediate; S–susceptible; AMC–amoxicillin/clavulanic acid; LEX–cephalexin; ENX–enrofloxacin; MBX–marbofloxacin; TET–tetracycline. Numbers inside brackets '()' indicate the total number of tested isolates for each antibiotic.

sulfamethoxazole, amoxicillin/clavulanic acid, and cephalexin. In contrast, _E. coli_ isolates from cats showed high proportions of non-susceptibility towards tetracycline (95.2%). Significant differences were observed between the _E. coli_ isolates from cats and dogs to tetracycline (95.2% versus 64.7%) (Table 3). It was observed that the frequency of MDR _E. coli_ isolates from cats and dogs were similar (70.8% versus 72.2%; $\chi^2 = 0.024$, $p = 0.877$).

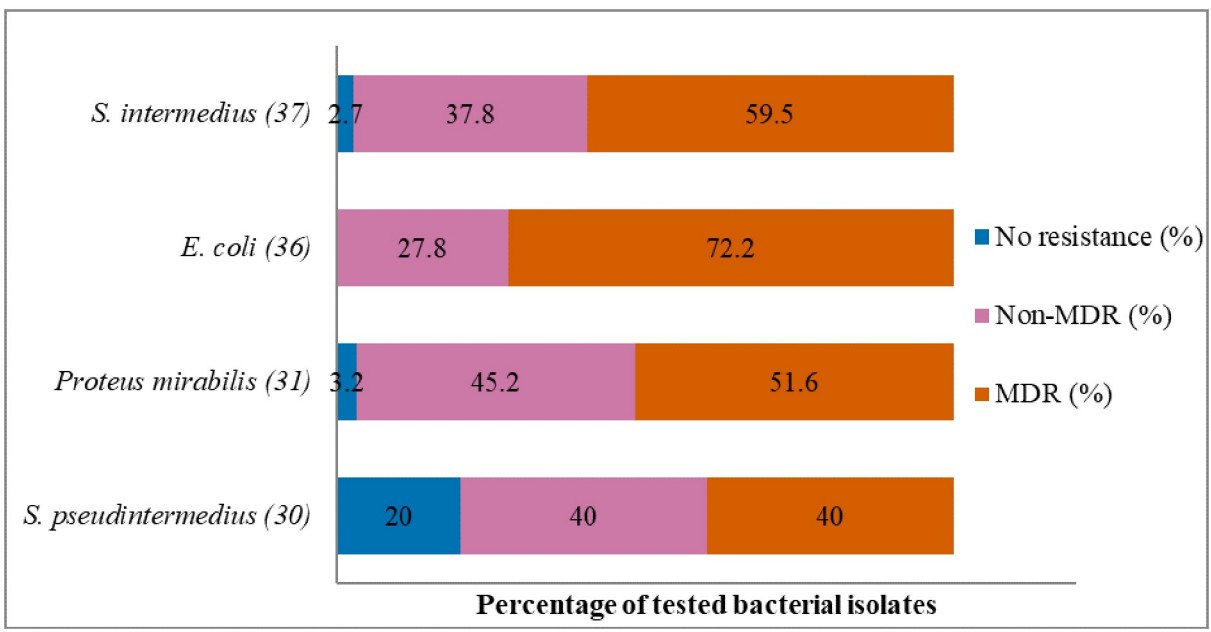

**Fig 11. Multidrug resistance of clinically important bacterial pathogens from diseased dogs between 2015 and 2017.** Numbers inside brackets '()' indicate number of isolates detected; those on bars indicate percentages per organism; Non-MDR–Resistant to 1 or 2 classes; MDR–multi-drug resistant, resistant to 3 or more classes.

**Table 3. Difference between *E. coli* isolates from diseased cats and dogs between 2015 and 2017.**

| Antimicrobial agents | Cat | | Dog |
|---|---|---|---|
| | Non-susceptible % (95% CI) | *p*-value | Non-susceptible % (95% CI) |
| Amoxicillin/clavulanic acid | 96.9 (89.2–99.6) | 0.338 | 91.2 (76.3–98.1) |
| Marbofloxacin | 64.1 (51.1–75.7) | 0.222 | 76.7 (57.7–90.1) |
| Cephalexin | 100 (92.6–100) | 0.368 | 96.4 (81.7.-99.9) |
| Enrofloxacin | 80 (65.4–90.4) | 0.624 | 84.4 (67.2–94.7) |
| Tetracycline* | 95.2 (76.2–99.9) | 0.031 | 64.7 (38.3–85.8) |
| Trimethoprim/sulfamethoxazole | 58.3 (27.7–84.8) | 0.371 | 83.3 (51.6–97.9) |
| Ceftriaxone | 42.9 (17.7–71.1) | NA | NA |
| Doxycycline | 100 (69.2–100) | NA | NA |
| Gentamicin | 90 (55.5–99.7) | NA | NA |

NA- Not analyzed (data with isolates less than 10 were not analyzed);

*Significant difference between cats and dogs, P < 0.05.

**Staphylococcus pseudintermedius.** The resistance patterns of *S. pseudintermedius* isolates to commonly used antimicrobial agents were similar between dogs and cats (Table 4). The frequency of MDR also shared similarities between *S. pseudintermedius* isolates from cats and dogs (40.3% versus 40%; $\chi^2 = 0.001$, $p = 1.000$).

## Discussion

AMR is an ongoing global public health challenge that negatively impacts animals, humans as well as environmental health. Data from veterinary diagnostic laboratories provide vital information about trends and patterns of AMR among disease-causing agents circulating in the local animal population. In this study, the analyses of the data supplies important snapshot information on the AMR situation to shape treatment regime and to guide policy on the use of antibiotics in pets [37–39], consequently prolonging the usefulness of commonly prescribed antibiotics.

Our previous study reported a very high AMR prevalence among major disease-causing bacteria in livestock [30]. In this study, we report that S. *pseudintermedius*, *E. coli*, *S. intermedius*, *P. mirabilis*, *K. pneumoniae*, and *S. canis*, which are the most common pathogens isolated

**Table 4. Differences between *S. pseudintermedius* isolates from diseased cats and dogs between 2015 and 2017.**

| Antimicrobial agents | Cat | | Dog |
|---|---|---|---|
| | Non-susceptible % (95% CI) | *p*-value | Non-susceptible % (95% CI) |
| Amoxicillin/clavulanic acid | 65.7 (53.1–76.8) | 0.517 | 58.6 (38.9–76.5) |
| Marbofloxacin | 76.6 (64.3–86.2) | 0.668 | 72.4 (52.8–87.3) |
| Cephalexin | 57.7 (42.2–72.3) | 0.560 | 50 (27.2–72.8) |
| Enrofloxacin | 83.6 (71.2–92.2) | 0.544 | 77 (56.4–91) |
| Tetracycline | 75 (47.6–92.7) | 1.000 | 81.8 (48.2–97.7) |
| Trimethoprim/sulfamethoxazole | 78.5 (49.2–95.3) | NA | NA |
| Gentamicin | 73.9 (51.6–89.8) | NA | NA |
| Amoxicillin | 66.6 (38.4–88.2) | NA | NA |
| Azithromycin | 100 (69.2–100) | NA | NA |

NA- Not Analyzed (data with isolates less than 10 were not analyzed)

from diseased cats and dogs (2015–2017), have comparatively high levels of resistance to commonly used antibiotics for therapy. The resistance pattern against antibiotics for these pathogens underscores increasing challenges in treating common infections in cats and dogs as these bacterial species have also been identified as most clinically relevant in pets elsewhere [40]. Unfortunately, there are very few local or regional AMR information on pathogens causing diseases in pets to enable a more geographically contextual discussion. Therefore comparisons were made based on other available literatures.

## Antimicrobial resistance of clinical isolates from cats

Overall, the trend did not change significantly over the study period for all the antibiotics tested. The majority of the *S. canis* isolates remained sensitive to amoxicillin/clavulanic acid. However, the increasing level of *S. canis* isolates grouped as 'intermediate' for amoxicillin/clavulanic acid from 13.3% in 2015 to 22.2% in 2017 is a note of concern. The level of susceptibility to this antibiotic is markedly higher than observed in a study by Awosile et al., [41], which reported a very low level (0.6%) of resistance from samples received over a ten-year study period at the Diagnostic Services Bacteriology Laboratory, Canada [41]. Additionally, in the study conducted by Ludwig et al., [42] across Europe, AMR was reported at 4% to amoxicillin/clavulanic acid over the entire study period.

Fluoroquinolones enrofloxacin and marbofloxacin are specific formulations for veterinary use [43, 44]. Marbofloxacin, a broad-spectrum antibiotic, is a drug of choice for the exclusive treatment of chronic bacterial infections affecting the urinary and respiratory tracts as well as skin and ear infections [45]. We observed high levels of resistance to enrofloxacin and marbofloxacin for *K. pneumoniae* (>80%) and *E. coli* (>40%) isolates recovered from cats. These findings corroborate with other similar studies conducted around the world. For instance, Marques et al., [46] reported a 72% resistance level of *K. pneumoniae* isolates recovered from urine samples of diseased cats and dogs in Lisbon. Other researchers report a lower resistance level (39.3%) of *E. coli* isolates against enrofloxacin in Poland [22]. In contrast, lower resistance levels (range: 0–5%) were observed for *Enterobacteriaceae* against marbofloxacin from stray cats between 2017 and 2018 in South Korea [47]. The observed high resistance level against fluoroquinolones antibiotics is not surprising as they are frequently prescribed in veterinary production and clinical practice [48]. Our finding should provide evidence for precautionary actions from the veterinary authority and other sectors to better strategize its usage to ensure the antibiotic's continuing efficacy. Moreover, the overuse of fluoroquinolones in the veterinary application has been linked to cross-resistance with other fluoroquinolones such as ciprofloxacin and norfloxacin used for human therapy [19]. These are listed as critically important antimicrobials for humans and are often used for treating human bacterial infections [19, 49].

High resistance levels for *S. pseudintermedius* isolates from cats were recorded against azithromycin (90%). Azithromycin is a broad-spectrum macrolide used for bacterial, rickettsial, and parasitic infections [50, 51] and has been used in both veterinary and human medicine [19]. The resistance to this antibiotic is not commonly reported in pet animals. However, multiple reports of high-level resistance of azithromycin in animals are available amongst pathogens of public health significance in livestock [52] and human-specific pathogens such as *Salmonella* typhi [53] and *Neiserria* [54]. The high level of resistance points to the over-prescription of azithromycin and therefore must be re-examined as the antibiotic is preferred for treatment against rickettsial and hemotropica parasites [55–57].

In Finland [37] and Australia [58], the resistance levels for gentamicin and trimethoprim/sulfamethoxazole (30.8%-44.8%) for *S. pseudintermedius* isolates from cats were slightly lower than the findings of this study for the two antibiotics (>50%). Antibiotic-resistant *S.*

*pseudintermedius* is an emerging veterinary clinical pathogen with some reports highlighting its increasing level of multi-drug resistance [59]. Although the zoonotic transmission of *S. pseudintermedius* remains rare [60], a few reports have linked infection from cats to their owners [61, 62].

### Antimicrobial resistance of clinical isolates from dogs

The resistance trends for cephalexin and tetracycline in *S. pseudintermedius* and marbofloxacin in *P. mirabilis* isolates from dogs significantly declined between 2015 and 2017. *P. mirabilis* isolates were highly resistant to tetracycline (100%) and cephalexin (79.2%) which is concordant with the findings documented in Beijing [63] and Granada [64] where the isolates from dogs were 100% and 61.7% resistant against tetracycline, respectively. Auwaerter [65] reported the high resistance to tetracycline and cephalosporins by *P. mirabilis* isolates from dogs to be a common finding. Our findings of 79.2% resistance against cephalexin are higher than that reported in Canada (48.9%) by Awosile et al., [41]. The frequency of prescription of this antibiotic in small animal clinics [66] has been linked to the increasing level of resistance worldwide. Our study shows that more than half of *S. intermedius* and *S. pseudintermedius* isolates were resistant against tetracycline and is thus consistent with the findings from Kalhoro et al., [67] in sick dogs in China. A lower resistance level (less than 30%) of tetracycline by *S. pseudintermedius* isolates was reported from healthy dogs in Canada [68] and Australia [69].

*E. coli* isolates showing higher resistance levels against cephalexin (82.1%) and amoxicillin/clavulanic acid (76.5%) recorded in this study are similar to those reported among ESBL-*E. coli* isolates against amoxicillin/clavulanic acid (86.8%) from dogs and cats at the Veterinary Diagnostic Laboratory, United States [70]. Similar findings were also documented from dogs in Canada [41], Europe [71], the United States [72], and New Zealand [73]. Furthermore, in the present study, the *E. coli* isolates from dogs show high resistance levels against enrofloxacin, marbofloxacin, tetracycline, and trimethoprim/sulfamethoxazole, thus signifying an impending difficulty of treating *E. coli* infection in local dogs.

### Comparison of antimicrobial resistance of clinical isolates from dogs and cats

*E. coli* and *S. pseudintermedius* are regarded as an excellent sentinel for AMR in a wide range of companion animals and as part of the their normal microbiota [1, 74]. These opportunistic bacteria can cause a wide range of infections. Hence, they are suitable candidates for comparing resistance profiles between cats and dogs. Our study shows interspecies variation in resistance patterns to tetracycline for *E. coli* isolates. *E. coli* isolates originating from cats had significantly higher resistance levels than those from dogs, which is consistent with the report from Australia [75]. However, more research is needed to explain this finding.

When we compare our findings on *E. coli* isolates from pets for the purpose of this study to that from local diseased livestock [30], we found that the level of resistance to tetracycline is very similar to that from diseased ruminants (cattle and goat) but significantly lower compared to that from diseased non-ruminants (chicken and pig). Our finding supports the conjecture relating to the key role of the environment and frequency of antibiotic usage in the trends of AMR. The finding for tetracycline resistance of *E. coli* is similar to that of diseased ruminants in Malaysia [30] that are raised extensively or semi-intensively with limited usage of antibiotics. Non-ruminants have a significantly higher level of resistance to tetracycline [30] as this antibiotic is one of the most commonly used as feed additives in intensive farming systems [76, 77]. In contrast, tetracycline is used only for treatment for conditions such as eye infection and bacterial gastrointestinal infection in cats and dogs [78].

## Multi-drug resistance of clinical isolates from dogs and cats

Several bacteria are shared between companion animals and humans and this creates opportunities for interspecies transmission of AMR-bacteria [1]. In the current study, isolates from diseased dogs and cats exhibited high levels of MDR with the majority of the isolates being resistant to 3–4 antimicrobial agents. For example, we reported a high frequency of MDR among *E. coli* isolates from diseased dogs (72.2%) and cats (70.8%). We found a similar level of MDR among *E. coli* isolates from diseased livestock (both ruminant and non-ruminants) in our previous study [30]. However, with the exception of a study in Poland [22], which reported a 66.8% MDR among *E. coli* isolates from sick pets, many reports documented lower MDR levels. In the northern part of Peninsular Malaysia by Shahaza et al., [79] reported MDR *E. coli* isolates in only 0.58% of *E. coli* isolated from pets. Similarly, in Portugal [39] and Canada [41], a relatively lower MDR level (range: 13%–32%) was found among clinical *E. coli* isolates from sick cats and dogs presented to the Veterinary Diagnostic Laboratory. The level of MDR of other pathogens such as *S. pseudintermedius* from dogs and cats (>40%) is similar to those reported from a convenient sample of dogs (47.5% MDR) in the USA [80] but lower than that reported in another local study on *S. pseudintermedius* (n = 23) isolates from stray and pet dogs and cats (100% MDR) [81]. In contrast, Australia [59], France [25], and the United Kingdom [82] reported MDR levels of between 4.7% and 20.7%. The variations in the MDR levels could reflect the choice or preferences of some antimicrobials used for the treatment of bacterial diseases in pets [83] and also on the frequency of antimicrobial usage in livestock production settings as well as in human medicine within a locality or geographic region [84].

## Limitations of the study

The present study shares similar inherent biases as stated by Hayer, et al., [85] about the study limitations using veterinary diagnostic laboratory data. The data used for this study are laboratory reports generated from diagnostics cases; therefore, the findings may not represent the AMR situation in the general pet population in peninsular Malaysia and may not reflect the overall picture of AMR levels among bacteria in healthy pets. Additionally, the samples received by the lab were from diseased pets which may have been treated with antibiotics prior to sample collection, thereby altering the antibiogram profiles. Unfortunately, the histories regarding their previous antimicrobial therapy were mostly not available. In addition, the bacteria isolated could be a part of the normal flora or contaminants rather than the disease-causing agent. Finally, the panels of antibiotics used in the analyses were not consistent because they were dependent on several factors such as the request by the clinician, availability of antibiotics in the laboratory, and the type of organism isolated.

## Conclusion

This study presents a snapshot of the AMR among important bacterial pathogens such as Staphylococci (*S. pseudintermedius* and *S. intermedius*), *E. coli*, *K. pneumoniae*, *P. mirabilis*, and *S. canis* recovered from diseased pets. A similar AMR pattern was observed for *S. pseudintermedius* and *E. coli* isolates recovered from sick cats and dogs, except for their resistance to tetracycline—where *E. coli* isolates from cats show higher AMR levels than those from dogs. The majority of the aforementioned bacterial isolates also demonstrated high levels of MDR, thereby undermining or complicating treatment options for bacterial infections in pets. Despite the limitation in inferencing the findings of this study to larger populations, the high MDR levels observed alludes to the growing challenges concerning the treatment of pets with common bacterial infections. The findings from this study provide important baseline

measurements for the ongoing AMR surveillance and provide a foundation to better frame the antimicrobial stewardship efforts in pets.

## Acknowledgments

We thank the technical staff of the Bacteriology Laboratory, Faculty of Veterinary Medicine, UPM Serdang, Malaysia for their time and for sharing knowledge from the inception to the end of this research project.

## Author Contributions

**Conceptualization:** Nurul Asyiqin Haulisah, Latiffah Hassan.

**Data curation:** Nurul Asyiqin Haulisah, Latiffah Hassan.

**Formal analysis:** Nurul Asyiqin Haulisah, Latiffah Hassan.

**Funding acquisition:** Latiffah Hassan.

**Investigation:** Nurul Asyiqin Haulisah, Latiffah Hassan.

**Methodology:** Nurul Asyiqin Haulisah, Latiffah Hassan.

**Project administration:** Nurul Asyiqin Haulisah, Latiffah Hassan.

**Resources:** Nurul Asyiqin Haulisah, Latiffah Hassan.

**Software:** Nurul Asyiqin Haulisah, Latiffah Hassan.

**Supervision:** Latiffah Hassan, Nur Indah Ahmad, Siti Khairani Bejo.

**Validation:** Nurul Asyiqin Haulisah, Latiffah Hassan.

**Visualization:** Nurul Asyiqin Haulisah, Latiffah Hassan, Saleh Mohammed Jajere.

**Writing – original draft:** Nurul Asyiqin Haulisah, Latiffah Hassan, Saleh Mohammed Jajere, Nur Indah Ahmad, Siti Khairani Bejo.

**Writing – review & editing:** Nurul Asyiqin Haulisah, Latiffah Hassan.

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
