## [Decision Letter · Decision Letter 0]

18 Aug 2022

PONE-D-22-09971High Prevalence of Antimicrobial Resistance and Multidrug Resistance Among Bacterial Isolates from Diseased Pets: Retrospective Laboratory Data (2015-2017)PLOS ONE

Dear Dr. Hassan,

Thank you for submitting your manuscript to PLOS ONE. After careful consideration, we feel that it has merit but does not fully meet PLOS ONE’s publication criteria as it currently stands. Therefore, we invite you to submit a revised version of the manuscript that addresses the points raised during the review process. As you will see, the referee asks for some details that should be adressed in a further version of your work. Along the same lines, I think it would be recomedable to include some information about the situation of AMR in wildlife (e.g., One Health,11,2020,100198) that is turning into a serious animal health issue.

We look forward to receiving your revised manuscript.

Kind regards,

Emmanuel Serrano, PhD

Academic Editor

PLOS ONE

Journal Requirements:

Reviewers' comments:

Reviewer's Responses to Questions

**Comments to the Author**

1. Is the manuscript technically sound, and do the data support the conclusions?

Reviewer #1: Yes

2. Has the statistical analysis been performed appropriately and rigorously? 

Reviewer #1: I Don't Know

3. Have the authors made all data underlying the findings in their manuscript fully available?

Reviewer #1: Yes

4. Is the manuscript presented in an intelligible fashion and written in standard English?

Reviewer #1: Yes

5. Review Comments to the Author

Reviewer #1: GENERAL COMMENTS

• Overall, this is an interesting and timely paper on the evidence on the emerging Antimicrobial Resistance in companion animals in Malaysia.

• That being said there are several areas for clarification and elaboration that would strengthen the study.

• Scope:

o Although the aim of the paper was to analyze a retrospective data between 2015 to 2017 among companion animals presented to veterinary hospital, a significant gap is the lack of inclusion of data between 2017- 2022 missing accurate information on the current picture of AMR in the target population.

o Given that the prevalence of AMR is highly likely to be low in household-based setting compared to hospital setting as the case here, the authors need to clarify why this pertinent data was excluded.

o Related to the veracity of the confounding factors in some circumstances, the reported prevalence may not be specific, hence the lower proportion of these than expected. To establish this as evidence, the authors might make a correlation between antimicrobial use (AMU) in companions animals vs AMR to identify pathways of the drivers of AMR in Malaysia. Comparison of AMR between different settings (hospital as the case here vs household would further strengthen the impact of the paper.

Abstract

• Rationale

o My concern is that the brief of the rationale and the existing gaps has NOT been be briefly described.

o The authors presented retrospective data retrieved between 2015 – 2017 and up to date data missing great proportion of the totality of the evidence after 2017 [5 years].

MATERIALS & METHODS

Source of the data

o The appended below sentence (see line 90-94) should be shifted to results section as it is not related to the methodology.

Fig 1 shows the distribution and percentages of bacterial species isolated during the study period. Escherichia coli was the most common isolate for Gram-negative bacteria followed by Proteus mirabilis and Klebsiella, whereas for Gram-positive bacteria, Staphylococcus pseudintermedius was the most common isolate followed by Streptococcus canis and Staphylococcus intermedius.

o Please clarify the following sentence:

‘For the purpose of this analysis, the isolates categorized as ‘intermediate’ and ‘resistant’ were grouped as ‘non-susceptible’.

• RESULTS

o Please define the acronym used in the fig 2-11. Please apply this change throughout.

o In Table 1: samples [postmortem, reproductive, urinary tract], please specify the nature of sample, and type of the organs rather than categorizing them as system-based classification.

o Please add a footnote to the table 1 and introduce the [others], etc.

o Please write as Fig 2-5 in line 145 and Table 3, 4 in line 293

o Rectify the footnote of Table 4; [*NA- should be written NOT Analyzed rather than Not Included to avoid confusion(see line 317).

• DISCUSSION

o The discussion section, even though should NOT be sub structured, remains superficial and not country specific. Comparisons are often made at global scale rather than regional and national levels missing crucial information about the epidemiology of AMR in the context of current research in Malaysia and the wider Southeast Asia.

o The author (s) should also discuss the strength of their study and how their manuscript contribute to the literature and fit within the context of current research or practice.

o

o The author (s) should also discuss the limitation of their study of using admitted cases as these that will overestimate of the prevalence of AMR, thus the call for a nationwide AMR prevalence study.

6. PLOS authors have the option to publish the peer review history of their article (what does this mean?). If published, this will include your full peer review and any attached files.

Reviewer #1: **Yes: **Abdinasir Yusuf Osman

---

## [Author Response · Author response to Decision Letter 0]

13 Sep 2022

Title: High Prevalence of Antimicrobial Resistance and Multidrug Resistance among Bacterial Isolates from Diseased Pets: Retrospective Laboratory Data (2015-2017)

Dear Reviewer, 

We appreciate the constructive comments from the reviewer and believe that these comments have improved the overall quality and readability of the manuscript. Below are the point-by-point responses to the comments of the reviewer. We sincerely hope that this manuscript is now more acceptable for further review and publication in this journal. 

NO. COMMENTS FROM THE REVIEWER RESPONSE AND CORRECTIONS MADE

 A. SCOPE 

1. Although the aim of the paper was to analyze a retrospective data between 2015 to 2017 among companion animals presented to veterinary hospital, a significant gap is the lack of inclusion of data between 2017-2022 missing accurate information on the current picture of AMR in the target population.

 This project was part of a Master research project which started in 2018, therefore was scoped for the previous three-year retrospective lab report (2015-2017). We appreciate the suggestion of expanding the period of study however given the timeline of the Master’s programme, this was not possible. 

However, since almost no information exists in the country and very few publications exists regionally or globally on the AMR of clinically important pathogens of small animals, we believe that this study has provided significant information on the overall picture of resistance pattern of pathogens that are important in causing illnesses in cats and dogs giving insight about the local trends of AMR at the national, regional, and global level.

2. Given that the prevalence of AMR is highly likely to be low in household-based setting compared to hospital setting as the case here, the authors need to clarify why this pertinent data was excluded.

 The overall aim of this research is to address the knowledge gap and describe the situation of the AMR among clinical isolates through a retrospective examination of available diagnostic data. Most cases presented were from veterinary health premises and animal facilities received by the accredited diagnostic bacteriology laboratory at the university. We have made it explicit that data from this study does not necessarily represent the prevalence of AMR from pets in households or general cats and dogs populations in Peninsular Malaysia.

3. Related to the veracity of the confounding factors in some circumstances, the reported prevalence may not be specific, hence the lower proportion of these than expected. To establish this as evidence, the authors might make a correlation between antimicrobial use (AMU) in companions animals vs AMR to identify pathways of the drivers of AMR in Malaysia. Comparison of AMR between different settings (hospital as the case here vs household would further strengthen the impact of the paper.

 Thank you for the comment. We fully recognized and acknowledged the inherent limitations of our findings and have addressed this in the original manuscript (Page 27). Unfortunately, the samples received by the diagnostic laboratory from pets were collected from diseased animals with incomplete history of antibiotic treatment(s). Thus, we cannot correlate between antimicrobial use (AMU) and antimicrobial resistance (AMR). In addition, as with many parts of the world, but especially reflective of the local and regional situation at the time of the study and remained true at present, the use of antibiotics in animals for both livestock and pets are poorly monitored. Monitoring and quantifying AMU is one of the strategy for the national AMR action plan for Malaysia that has been extended into the next action plan 2022-2027 due to the complexity and challenges of acquiring this type of information. 

 B. ABSTRACT (Rationale) 

4. My concern is that the brief of the rationale and the existing gaps has NOT been being briefly described.

 See No. 1

5. The authors presented retrospective data retrieved between 2015 – 2017 and up to date data missing great proportion of the totality of the evidence after 2017 [5 years].

 See No. 1

 C. MATERIALS AND METHODS (Source of the data) 

6. The appended below sentence (see line 90-94) should be shifted to results section as it is not related to the methodology.

• Fig 1 shows the distribution … followed by Streptococcus canis and Staphylococcus intermedius.

 Figure 1 has been shifted to the Results section in page 7.

 Please clarify the following sentence:

• ‘For the purpose of this analysis, the isolates categorized as ‘intermediate’ and ‘resistant’ were grouped as ‘non-susceptible’. For the purpose of statistical analysis (Chi-square test), those isolates categorized as ‘intermediate’ from antibiotic susceptibility testing were not excluded from the study, instead were combined and grouped together with ‘resistant’ and named ‘non-susceptible’. 

 D. RESULTS 

7. Please define the acronym used in the fig 2-11. Please apply this change throughout.

 The acronyms were readily presented in the original manuscript. 

8. In Table 1: samples [postmortem, reproductive, urinary tract], please specify the nature of sample, and type of the organs rather than categorizing them as system based classification.

 The samples of the diseased animals presented to the bacteriology laboratory from various types of samples were combined and categorized as system-based increase data robustness for analysis because the number of isolates per type of samples varies widely and can be low.

9. Please add a footnote to the table 1 and introduce the [others], etc.

 Footnote for table 1 and 2 has been added in page 9 and 14

10. Please write as Fig 2-5 in line 145 and Table 3, 4 in line 293.

 Corrected in page 9 and 18

11. Rectify the footnote of Table 4; [*NA- should be written NOT Analyzed rather than Not Included to avoid confusion (see line 317).

 Corrected in page 19 and 20

 E. DISCUSSION 

12. The discussion section, even though should NOT be sub structured, remains superficial and not country specific. Comparisons are often made at global scaler rather than regional and national levels missing crucial information about the epidemiology of AMR in the context of current research in Malaysia and the wider Southeast Asia.

 Thank you for the comment and we acknowledge the limitation of available literature to discuss the findings of this study. Research and publication of AMR in the veterinary context and especially focusing on pathogens in pets is not extensive in Malaysia and in the region. We have addressed this issue in the Introduction and added a few lines in the Discussion (Page 21 Linen 383-385). 

In Malaysia, there is only one publication by Shahaza et al., [79] which was included in the discussion part of this manuscript, that presented the analysis of E. coli of clinical isolates from various species. There is no publication about clinical isolates in pets at the regional level which was why most of the references and comparisons were from more extended populations of pets in Europe, Canada and Australia. 

13. The author (s) should also discuss the strength of their study and how their manuscript contribute to the literature and fit within the context of current research or practice.

 The strength of this study was discussed in Introduction, page 4, line 63 to 68. 

14. The author (s) should also discuss the limitation of their study of using admitted cases as these that will overestimate of the prevalence of AMR, thus the call for a nationwide AMR prevalence study.

 The limitation of the study was discussed in page 27.

---

## [Decision Letter · Decision Letter 1]

10 Oct 2022

PONE-D-22-09971R1High Prevalence of Antimicrobial Resistance and Multidrug Resistance among Bacterial Isolates from Diseased Pets: Retrospective Laboratory Data (2015-2017)PLOS ONE

Dear Dr. Hassan,

Thank you for submitting your manuscript to PLOS ONE. After careful consideration, we feel that it has merit but does not fully meet PLOS ONE’s publication criteria as it currently stands. Therefore, we invite you to submit a revised version of the manuscript that addresses the points raised during the review process. Please submit your revised manuscript by Nov 24 2022 11:59PM. If you will need more time than this to complete your revisions, please reply to this message or contact the journal office at plosone@plos.org. Please include the following items when submitting your revised manuscript:A rebuttal letter that responds to each point raised by the academic editor and reviewer(s). You should upload this letter as a separate file labeled 'Response to Reviewers'.A marked-up copy of your manuscript that highlights changes made to the original version. You should upload this as a separate file labeled 'Revised Manuscript with Track Changes'.An unmarked version of your revised paper without tracked changes. You should upload this as a separate file labeled 'Manuscript'.If applicable, we recommend that you deposit your laboratory protocols in protocols.io to enhance the reproducibility of your results. Protocols.io assigns your protocol its own identifier (DOI) so that it can be cited independently in the future. For instructions see: https://journals.plos.org/plosone/s/submission-guidelines#loc-laboratory-protocols. Additionally, PLOS ONE offers an option for publishing peer-reviewed Lab Protocol articles, which describe protocols hosted on protocols.io. Read more information on sharing protocols at https://plos.org/protocols?utm_medium=editorial-email&utm_source=authorletters&utm_campaign=protocols.

We look forward to receiving your revised manuscript.

Kind regards,

Emmanuel Serrano, PhD

Academic Editor

PLOS ONE

Journal Requirements:

Reviewers' comments:

Reviewer's Responses to Questions

**Comments to the Author**

1. If the authors have adequately addressed your comments raised in a previous round of review and you feel that this manuscript is now acceptable for publication, you may indicate that here to bypass the “Comments to the Author” section, enter your conflict of interest statement in the “Confidential to Editor” section, and submit your "Accept" recommendation.

Reviewer #1: All comments have been addressed

2. Is the manuscript technically sound, and do the data support the conclusions?

Reviewer #1: Yes

3. Has the statistical analysis been performed appropriately and rigorously? 

Reviewer #1: Yes

4. Have the authors made all data underlying the findings in their manuscript fully available?

Reviewer #1: Yes

5. Is the manuscript presented in an intelligible fashion and written in standard English?

Reviewer #1: Yes

6. Review Comments to the Author

Reviewer #1: 1) The presented figures as a whole show poor quality and thus re-plotting probably using more sophisticated software to enhance their quality is highly recommended

2) The strength of the study should be written [re-highlighted] in the discussion section.

7. PLOS authors have the option to publish the peer review history of their article (what does this mean?). If published, this will include your full peer review and any attached files.

Reviewer #1: **Yes: **Abdinasir Yusuf Osman

---

## [Author Response · Author response to Decision Letter 1]

31 Oct 2022

Title: High Prevalence of Antimicrobial Resistance and Multidrug Resistance among Bacterial Isolates from Diseased Pets: Retrospective Laboratory Data (2015-2017)

Dear Reviewer, 

We appreciate the constructive comments from the reviewer and believe that these comments have improved the overall quality and readability of the manuscript. Below are the point-by-point responses to the comments of the reviewer. We sincerely hope that this manuscript is now more acceptable for further review and publication in this journal. 

NO. COMMENTS FROM THE REVIEWER RESPONSE AND CORRECTIONS MADE

 A. REFERENCES 

 1) In text citation for:

• [1] (Page 2, Line 35)

• [1] (Page 26, Line 464)

• [22–24] (Page 4, Line 63) 

• [1,74] (Page 25, Line 437) has been updated to the latest version.

2) Reference for: 

24. Moyaert H, de Jong A, Simjee S, Rose M, Youala M, El Garch F, et al. Survey of antimicrobial susceptibility of bacterial pathogens isolated from dogs and cats with respiratory tract infections in Europe: ComPath results. J. Appl. Microbiol. 2019 April; 127(1): 29-46. https://doi.org/10.1111/jam.14274

has been revised to 

24. Moyaert H, de Jong A, Simjee S, Rose M, Youala M, El Garch F, et al. Erratum: Survey of antimicrobial susceptibility of bacterial pathogens isolated from dogs and cats with respiratory tract infections in Europe: ComPath results. J. Appl. Microbiol. 2019 November; 127(5): 1594. https://doi.org/10.1111/jam.14420

3) Statement for:

• (Page 2, Line 33-35): ……… isolates in pets is an emerging issue of veterinary and public health significance [1] has been revised to ……… isolates such as MRSA and MDR E. coli and Klebsiella pneumoniae in pets is an emerging issue of veterinary and public health significance [1]. 

• (Page 6, Line 114-116): ……… as ‘multi-drug resistant’ if it were resistant to at least three or more classes of antibiotics [36] has been revised to ……… as ‘multi-drug resistant’ if it were resistant to at least one agent in three or more classes of antibiotics [36].

• (Page 25, Line 436-437): ……… wide range of pets and as part of the normal microbiota of pets [1,74] has been revised to ……… wide range of companion animals and as part of the their normal microbiota [1,74]. 

• (Page 26, Line 463-464): ……… between pets and humans and this creates opportunities for interspecies transmission of AMR-bacteria [1] has been revised to ……… between companion animals and humans and this creates opportunities for interspecies transmission of AMR-bacteria [1]. 

 B. FIGURES 

2. The presented figures as a whole show poor quality and thus re-plotting probably using more sophisticated software to enhance their quality is highly recommended Figures have been updated to the good quality using Preflight Analysis and Conversion Engine (PACE) digital diagnostic tool.

 C. STRENGTH OF THE STUDY 

3. The strength of the study should be written [re-highlighted] in the discussion section. The strength of the study has been highlighted in the discussion section (Page 20-21, Line 343-350)

---

## [Editor Report · Decision Letter 2]

2 Nov 2022

High Prevalence of Antimicrobial Resistance and Multidrug Resistance among Bacterial Isolates from Diseased Pets: Retrospective Laboratory Data (2015-2017)

PONE-D-22-09971R2

Dear Dr. Hassan,

We’re pleased to inform you that your manuscript has been judged scientifically suitable for publication and will be formally accepted for publication once it meets all outstanding technical requirements.

Kind regards,

Emmanuel Serrano, PhD

Academic Editor

PLOS ONE
---

## [Editor Report · Acceptance letter]

10 Nov 2022

PONE-D-22-09971R2 

High Prevalence of Antimicrobial Resistance and Multidrug Resistance among Bacterial Isolates from Diseased Pets: Retrospective Laboratory Data (2015-2017) 

Dear Dr. Hassan:

I'm pleased to inform you that your manuscript has been deemed suitable for publication in PLOS ONE. Congratulations! Your manuscript is now with our production department. 

Kind regards, 

on behalf of

Dr. Emmanuel Serrano 

Academic Editor

PLOS ONE